# Active vaccine safety surveillance: Experience from a prospective cohort event monitoring study of COVID-19 vaccines in Kenya

**Don B. Odhiambo**[1]*, **Donald Akech**[1], **Boniface Karia**[1], **Makobu Kimani**[1],
**Samuel Sang**[1], **Antipa Sigilai**[1,2], **Shirine Voller**[1,2], **Christine Mataza**[3],
**David Mang'ong'o**[3], **Rose Jalang'o**[4], **Martha Mandale**[5], **Anthony O. Etyang**[1],
**John Anthony Gerard Scott**[1,2], **Ambrose Agweyu**[1,2], **Eunice Wangeci Kagucia**[1]

**1** KEMRI-Wellcome Trust Research Programme, Kilifi, Kenya, **2** London School of Hygiene and Tropical Medicine, London, United Kingdom, **3** Department of Health, County Government of Kilifi, Kilifi County, Kilifi, Kenya, **4** National Vaccination and Immunization Programme, Ministry of Health, Government of Kenya, Nairobi, Kenya, **5** Pharmacy and Poisons Board, Ministry of Health, Government of Kenya, Nairobi, Kenya

* dbrian@kemri-wellcome.org

## Abstract

Although active vaccine safety surveillance (VSS) can complement passive VSS while overcoming the inherent limitations of spontaneous safety monitoring, it remains rare in sub-Saharan Africa. We conducted post-authorization active VSS of COVID-19 vaccines in Kilifi, Kenya using a cohort event monitoring study design. Participants were followed weekly over 13 weeks for adverse events. A subset was followed daily for one week for solicited systemic reactogenicity events (chills, fatigue, fever, headache, joint pain, malaise, muscle aches, nausea). The daily prevalence of reactogenicity events was compared to the 3-day pre-vaccine average using McNemar's test. The association of baseline characteristics with reactogenicity events was assessed using logistic regression. Between 28th September 2022 and 30th June 2023, 2,440 participants were enrolled into the cohort; 1,000 systematically sampled participants were included in the reactogenicity sub-study. Most were aged 17–39 years (1683; 69.0%) and were female (1895; 77.7%); 535 (28.2%) female participants were pregnant. The three most frequently reported reactogenicity events were fatigue (422; 44.1%), headache (370; 38.7%), and malaise (346; 36.2%); the proportion of severe events ranged from 2.3% (22; nausea) to 5.0% (48; malaise). Except for headache, the prevalence of systemic reactogenicity events was significantly higher in the first two days post-vaccination than pre-vaccination (p-values <0.05). The odds of reactogenicity events were higher among non-pregnant women (adjusted odds ratio [aOR] 1.81; 95% CI 1.28-2.55) and pregnant women (aOR 1.69; 1.03-2.78) than among men, and higher among Johnson & Johnson (aOR 2.05; 1.40-3.00) and Moderna (aOR 4.19; 2.34-7.51) vaccine recipients than among Pfizer vaccine recipients. The prevalence of pregnancy complications was 2.6% (95% CI

**Data availability statement:** The deidentified data set and other supplementary materials have been published on the Harvard dataverse server: DOI: https://doi.org/10.7910/DVN/15QZIF. This work is licensed under a Creative Commons Attribution 4.0 International (CC BY 4.0) license, which permits unrestricted use, distribution, and reproduction in any medium, provided the original work is properly cited. To view a copy of this license, visit https://creativecommons.org/licenses/by/4.0/.

**Funding:** This study was supported by the Foreign, Commonwealth & Development Office (FCDO) in the form of an institutional grant awarded to EWK (Project Number 300708-159) and the University of Oxford on behalf of the KEMRI-Wellcome Trust Research Programme in the form of a salary for DBO, DA, BK, AS, and EWK. The specific roles of these authors are articulated in the 'author contributions' section. The funders had no role in study design, data collection and analysis, decision to publish, or preparation of the manuscript.

**Competing interests:** A.A. reports institutional grants from the Gates Foundation, National Institute for Health and Care Research (NIHR), Wellcome Trust, Medical Research Council and FCDO, meeting/ travel support from the Gates Foundation, and an advisory role in a World Health Organization committee. E.W.K reports institutional grants from the Gates Foundation and FCDO, and meeting/ travel support from the Gates Foundation. The other authors have no interests to declare.

1.4-3.5%) against a background prevalence of 3–49%. Reactogenicity events following COVID-19 vaccination were generally non-severe and transient. There was no elevated risk of pregnancy-related complications. Addressing operational barriers is essential for enhancing the utility and feasibility of future active VSS.

## Introduction

Vaccine safety monitoring in sub-Saharan Africa relies on passive surveillance systems, which require fewer resources than active surveillance, representing a feasible approach for national pharmacovigilance [1,2]. However, the quality and utility of data from passive vaccine safety surveillance (VSS) is limited by reporting bias, incomplete reporting, and a lack of accurate denominator data [3]. Yet, reliable safety data are needed to rapidly respond to vaccine safety concerns when they arise. Active VSS can be deployed at critical junctures, such as at new vaccine introduction and/ or in response to emerging vaccine safety concerns. In this way, active VSS can complement existing passive VSS programs, generating the evidence needed to ensure vaccine safety and maintain public trust in vaccines.

The COVID-19 pandemic created a public health need for vaccines to be deployed as part of disease control efforts. To contain the pandemic, the World Health Organization (WHO) and other global actors advocated for a rapid and massive vaccine roll out globally. Beginning in November 2020, COVID-19 vaccines received emergency use approvals to support their rapid deployment globally [4]. The Kenya Ministry of Health began implementation of the national COVID-19 vaccination program in early March 2021 in line with WHO recommendations, first targeting priority groups such as health workers, security personnel, teachers, and vulnerable individuals >58 years of age, followed by all adults and, later, children aged 12–17 years [5,6]. Through the COVID-19 Vaccine Global Access (COVAX) facility and bilateral agreements, a total of five different products were made available in Kenya within the national COVID-19 vaccination program: Oxford/ AstraZeneca (Covishield), Pfizer-BioNTech (Cominarty/ BNT162b2), Johnson & Johnson/ Janssen (Ad26.COV2.S), Moderna (Spikevax/ mRNA-1273),and Sinopharm (BBIBP-CorV). There was a brief and limited use of Sputnik V within the private sector in March-April 2021 [7]. By March 2023, approximately 14 million persons among adults aged ≥18 years and children aged 12–17 years in Kenya had received at least one dose of a COVID-19 vaccine, translating to 27 persons vaccinated with at least one dose per 100 [8].

COVID-19 VSS in high-income countries revealed an association between COVID-19 vaccines and rare though serious adverse events following immunization (AEFI) such as thrombosis and thrombocytopenia syndrome (TTS) following receipt of vectored vaccines [9] and myocarditis/ pericarditis following receipt of mRNA COVID-19 vaccines [10–12]. COVID-19 vaccine safety evaluations in low- and middle-income settings were needed to provide evidence from populations in alternative epidemiologic contexts. In Kenya, serological surveillance revealed that up to half of the general population had been infected with severe acute respiratory syndrome

coronavirus 2 (SARS-CoV-2) by the time the national COVID-19 vaccination program was rolled out, [13] different from high-income settings were COVID-19 seroprevalence was comparatively lower at vaccine introduction. In addition, relative to most high-income COVID-19 vaccination programs, the program in Kenya was complex, characterized by various permutations of multiple products and schedules.

VSS in Kenya is conducted jointly by the Pharmacy and Poisons Board (PPB), which is the national regulatory authority, and the National Vaccines and Immunization Program (NVIP). Healthcare practitioners and members of the public can submit spontaneous reports of AEFI to the PPB using paper forms or to the PPB's Pharmacovigilance Electronic Reporting System (PvERS) via a web-based application, mobile phone application, or mobile phone Unstructured Supplementary Service Data (USSD) code [14].

We undertook an active VSS study to generate evidence on the post-authorization safety of COVID-19 vaccines in Kenya. As a secondary objective, we evaluated the feasibility of following up vaccinated individuals using short message service (SMS).

## Methods

### Ethics statement

All participants provided written informed consent prior to study enrollment. The research protocol was approved by the Kenya Medical Research Institute Scientific and Ethics Review Unit (#4486), the Kenya National Commission for Science Technology & Innovation (NACOSTI/P/22/19448), the Oxford Tropical Research Ethics Committee (#28–22) and the London School of Hygiene & Tropical Medicine Research Ethics Committee (#28013). The research protocol was aligned with the WHO protocol for CEM for safety signal detection after vaccination with COVID-19 vaccine [15].

### Study design and participants

We undertook active safety surveillance study of COVID-19 vaccines using a cohort event monitoring (CEM) design. The study was conducted in Kilifi North, Kilifi South and Ganze sub-counties of Kilifi County, Kenya. Eligible individuals who had received a routine COVID-19 vaccination at the following seven vaccination centers were enrolled: Kilifi County Referral Hospital, Mnarani Dispensary, Mtwapa Sub-County Hospital, Bamba Sub-County Hospital, Gede Sub-County Hospital, Vipingo Health Centre, and Kizingo Dispensary (Fig 1). Enrolment into the study began on 28th September 2022 and ended on 31st March 2023.

Individuals who were vaccinated with any dose of any COVID-19 vaccine brand administered at the vaccination sites within the study area and who were available for follow-up duration were eligible for enrollment after providing written informed consent and a phone number through which they would be contacted during follow up. Unemancipated persons aged <18 years were excluded from the study as the study assent form required an amendment which was approved after enrollment was complete. Former participants could be re-screened and re-enrolled if they received an additional COVID-19 vaccine dose during the study period in line with the WHO protocol for CEM safety signal detection which allows follow-up of multiple doses per participant [15].

We aimed to enroll at least 10,000 participants. This sample size was sufficient to observe ≥1 event within 42 days for events occurring at a rate of ≥261 per 100,000 person years [16]. As such, if no event was observed, this sample size was sufficient to rule out, for example, a ≥ 4-fold increase in the risk of thrombosis (background rate of 80 per 100,000), [17] a ≥ 2-fold increase in the risk of acute respiratory distress syndrome (background rate of 39–193 per 100,000), [18,19] or a ≥ 1.5-fold increase in the risk of acute aseptic arthritis (background rate of 100–1600 per 100,000; [20,21] S1 Table). We also aimed to assign 1,000 participants into a sub-study evaluating reactogenicity events. This sample size was sufficient to yield reasonable margins of error across a wide range of potential prevalence estimates (S2 Table); for example, assuming a prevalence of 50%, the width of the exact binomial 95% confidence interval would be ± 3.1%.

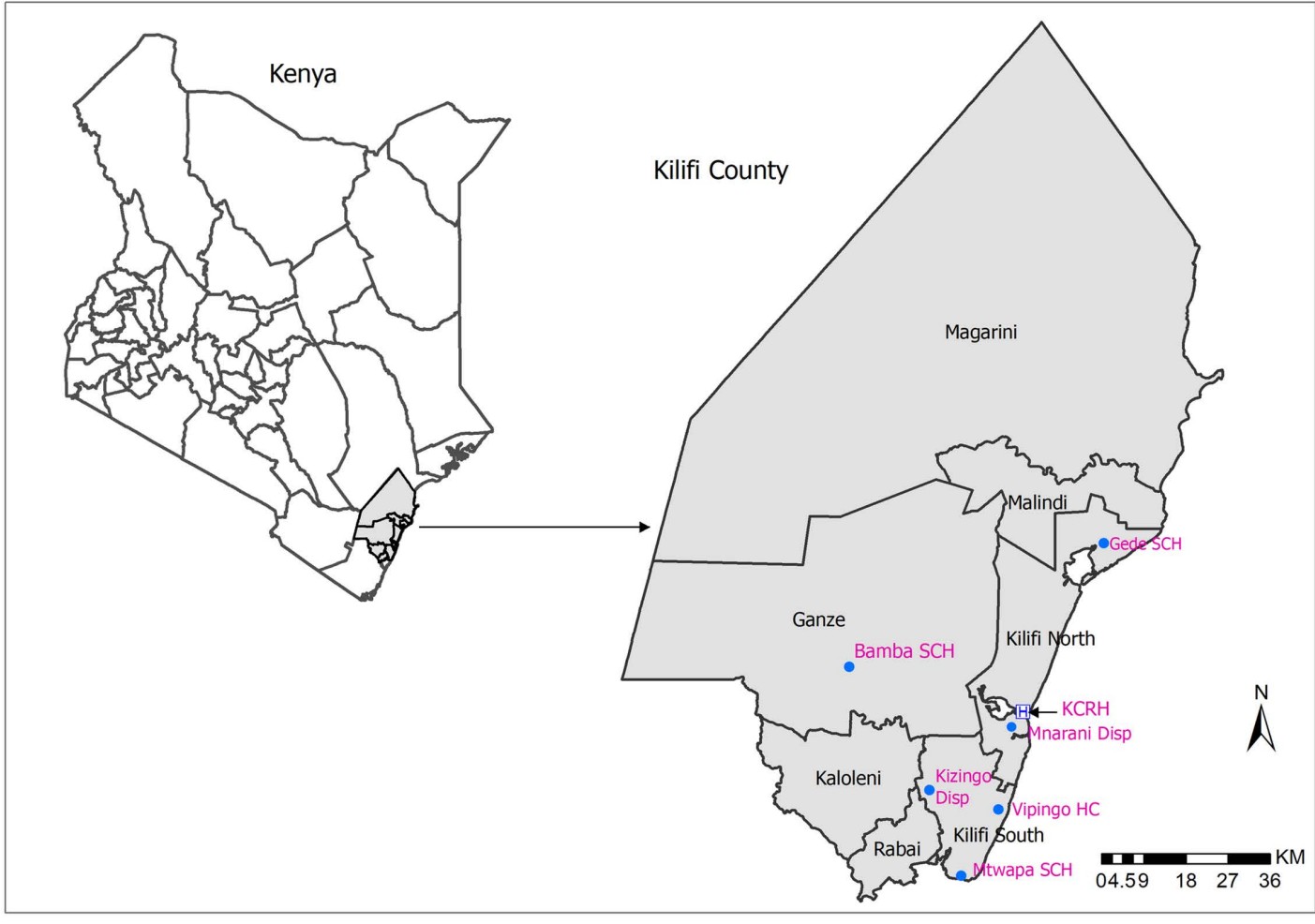

**Fig 1. Study site.** Map of Kilifi County, Kenya showing the various study vaccination sites. The base map was derived from Kenya administrative boundary shapefiles available at https://data.humdata.org/dataset/cod-ab-ken, provided by the Kenya National Bureau of Statistics under a CC BY 4.0 license. Map created using ArcGIS Desktop 10.8.2 (Esri, Redlands, CA, USA).

## Procedures

Eligible individuals were enrolled immediately after vaccination by study staff stationed at the vaccination facilities. The vaccinated individuals were enrolled consecutively provided they had met the eligibility criteria. Study staff collected the enrolled participants' baseline information which included sociodemographic characteristics, medical history (including self-reported pregnancy status), vaccination details and contact information (S1 Text). Beginning with the first set of study enrollees, participants were assigned to the reactogenicity sub-study in blocks of 50 participants with the subsequent block assigned when follow-up was complete for the previous block. This was continued until the target sample size of 1,000 was achieved. Enrollment into the reactogenicity sub-study was staggered to balance the workload for the study staff conducting follow-up for reactogenicity events. At enrolment, participants in the reactogenicity subset were further asked to retrospectively report any systemic symptoms they had experienced in the three days prior, excluding the date of enrolment.

Participants in the reactogenicity subset were followed up daily from Day 0 (day of vaccination) to Day 7 post-vaccination for solicited systemic reactogenicity events, i.e., fever, nausea, malaise, chills, headache, joint pain, muscle aches, and fatigue. All participants were followed up weekly up to 13 weeks for hospitalizations and any other post-vaccination events.

Study participants were followed up through telephone calls and/or short message service (SMS). The use of SMS-based follow-up was piloted to evaluate whether it may present a feasible alternative approach for participant follow-up, minimizing the human-resource needs associated with follow-up via phone call. The information gathered during phone call based follow up was captured using an online questionnaire by study staff while SMS-based responses were captured automatically using an online application. Online follow up questionnaires which expired before the participants responded were considered as missed questionnaires. Study staff made a total of three phone call or SMS contact attempts (on separate days) to reach participants with missed questionnaires. Unresponsive participants were declared lost to follow up if these contact attempts were unsuccessful.

Reactogenicity events were reported to the Kenya PPB by study staff via the online national adverse event reporting system, PvERS. Study staff referred serious AEFI and events of public concern to the respective Sub-County vaccine AEFI team for their evaluation and response as per the Ministry of Health guidelines. Causality assessment was not undertaken as part of the study; any AEFI reports submitted to the national pharmacovigilance system and needing causality assessment would be reviewed by the National Vaccines Safety Advisory Committee (NVSAC). Monthly safety surveillance summaries were provided to the NVIP and PPB to inform timely decision-making.

A total of 90 non-reactogenicity participants aged 18–45 years, who owned mobile phones, were selected through convenience sampling to participate in a SMS sub-study to determine the feasibility of using technology-based methods to carry out active community-based VSS in a resource-limited setting. They were followed up weekly (week 1–13) using a SMS based platform to gather information on post-vaccination events. Each participant received one to seven text messages during each weekly follow-up dependent on their responses. The text messages were sent out sequentially based on the response received from the preceding message. Participants who responded appropriately to all the text messages received a total of seven messages. Participants who did not respond at all received up to two reminder text messages within 24 hours requesting them to respond. Responses which did not feature in the predefined list of options were considered invalid (wrong responses). Participants who responded partially, who provided invalid responses or who did not respond at all were contacted through a telephone call to gather post-vaccination information. The proportion of each category of response was then determined.

## Statistical analysis

The primary outcome was the proportion of participants in the reactogenicity subset reporting solicited adverse events within the first 7 days after vaccination. The severity of these systemic events was also determined based on their ability to interfere with the normal daily activities of the participants. Mild events did not interfere with the normal daily activities of participants. Moderate events somewhat interfered with the normal daily activities of participants. Severe events were considerable and prevented the normal daily activities of participants. Our secondary outcome was the number of participants reporting serious AEFI, i.e., events that resulted into overnight hospitalization, death, persistent or significant disability, congenital defects or were life-threatening during the follow up period.

We used descriptive statistics to describe the baseline sociodemographic characteristics of the participants, the distribution of adverse events following immunization and the SMS response rates. We used the frequency of systemic reactogenicity events in the three days prior to vaccination to approximate the background rates of reactogenicity events and compared them against the daily frequency of post-vaccination systemic reactogenicity events using McNemar's test as the observations were not independent. We performed logistic regression to determine the association between age, sex, vaccine dose, vaccine brand, pregnancy status and comorbidities with any solicited systemic reactogenicity event as well

as with each solicited systemic reactogenicity event. All variables were included as covariates in the regression models to control for potential confounding.

Analyses were conducted using Stata (versions 15.0 & 17.0, StataCorp, College Station, TX).

## Results

### Baseline characteristics

A total of 3070 individuals who had been vaccinated at the various enrollment sites during the recruitment period were screened, out of which 2440 participants were enrolled (enrolment rate of 79.5%). Reasons for non-enrollment are indicated in Fig 2. A total of 1000 participants were enrolled into the reactogenicity sub-study (Fig 2). One participant was re-enrolled thereby being followed up over two COVID-19 vaccine doses; they were not in the reactogenicity subset for either follow-up. The follow-up period began immediately after the enrolment of the first participant and ended on 30th June 2023.

The baseline sociodemographic characteristics of the enrolled participants are summarized in Table 1 (the participant followed up over two vaccine doses is represented once). Detailed stratification of the vaccines received by the brand and

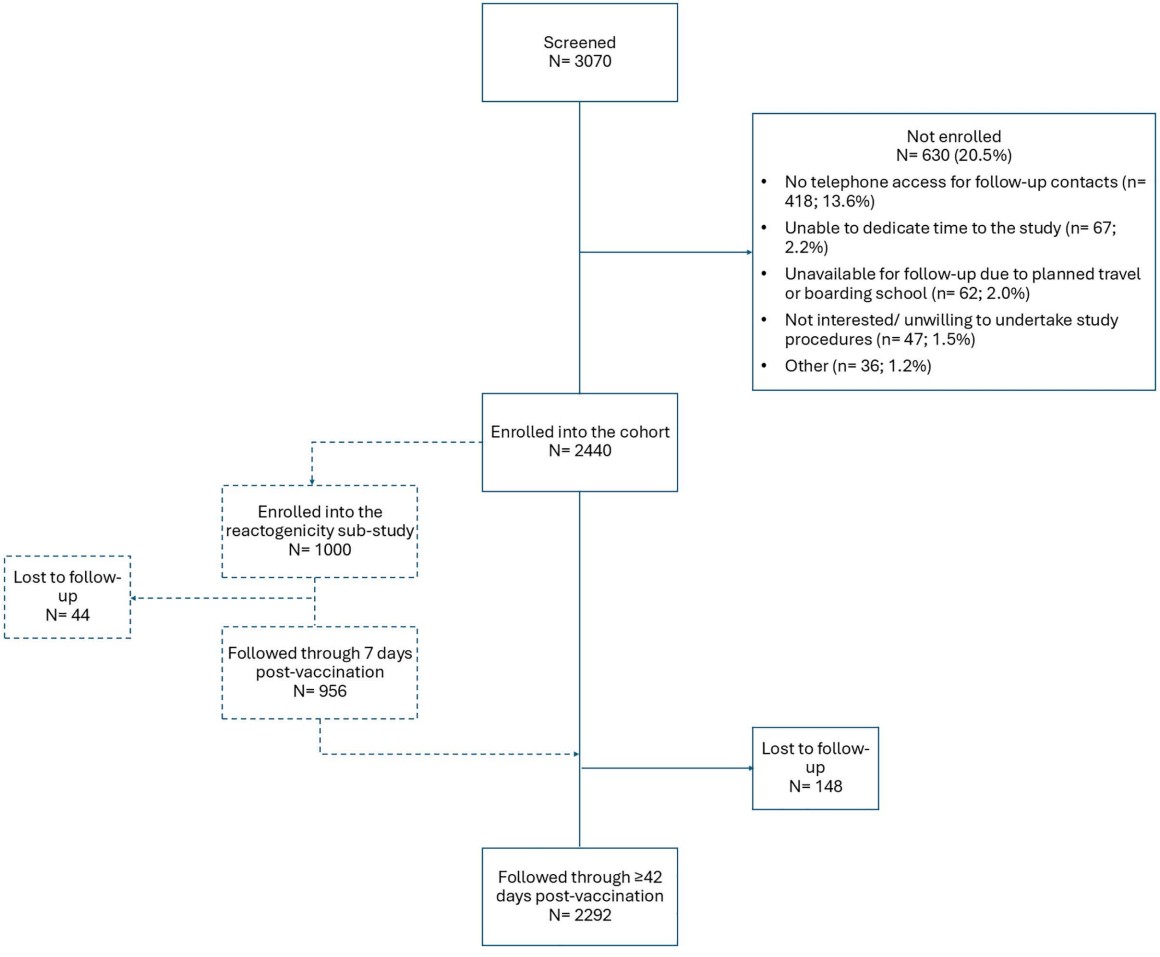

**Fig 2. Study participant flow diagram.**

**Table 1. Baseline sociodemographic characteristics of the enrolled participants.**

| Baseline sociodemographic characteristics [a] | | Reactogenicity subset N = 1000 n (%) | Non-reactogenicity subset N = 1439 n (%) | Overall N = 2439 n (%) |
|---|---|---|---|---|
| Age | 17-39yrs | 701 (70.1) | 982 (68.2) | 1683 (69.0) |
| | 40-59yrs | 228 (22.8) | 368 (25.6) | 596 (24.4) |
| | 60+yrs | 71 (7.1) | 89 (6.2) | 160 (6.6) |
| Sex | Male | 229 (22.9) | 316 (22.0) | 545 (22.3) |
| | Female | 771 (77.1) | 1123 (78.0) | 1894 (77.7) |
| Comorbidity | No | 721 (72.1) | 921 (64.0) | 1642 (67.3) |
| | Chronic respiratory disease or asthma | 40 (4.0) | 46 (3.2) | 86 (3.5) |
| | Chronic heart disease | 10 (1.0) | 33 (2.3) | 43 (1.8) |
| | Chronic renal disease | 3 (0.3) | 2 (0.1) | 5 (0.2) |
| | Diabetes | 6 (0.6) | 6 (0.4) | 12 (0.5) |
| | Immunocompromised/ immunosuppressed | 187 (18.7) | 396 (27.5) | 583 (23.9) |
| | Allergy | 33 (3.3) | 35 (2.4) | 68 (2.8) |
| Pregnant | No | 551 (71.5) | 808 (72.0) | 1359 (71.8) |
| | Yes | 220 (28.5) | 315 (28.0) | 535 (28.2) |
| Pre-vaccination systemic events [b] | None | 665 (66.5) | .. | .. |
| | Yes | 335 (33.5) | .. | .. |
| Previous COVID-19 disease | No | 979 (97.9) | 1418 (98.5) | 2397 (98.3) |
| | Yes, LC | 12 (1.2) | 10 (0.7) | 22 (0.9) |
| | Probable but not LC | 9 (0.9) | 11 (0.8) | 20 (0.8) |
| History of reaction to any vaccination | No | 930 (93.0) | 1293 (89.9) | 2223 (91.1) |
| | Yes | 70 (7.0) | 146 (10.1) | 216 (8.9) |
| Product mixing [c] | No | 136 (34.2) | 245 (40.2) | 381 (37.8) |
| | Yes | 262 (65.8) | 365 (59.8) | 627 (62.2) |
| Vaccine Brand | Pfizer | 381 (38.1) | 499 (34.7) | 880 (36.1) |
| | Johnson & Johnson | 513 (51.3) | 654 (45.4) | 1167 (47.8) |
| | Moderna | 106 (10.6) | 286 (19.9) | 392 (16.1) |
| Vaccine Dose | 1st | 602 (60.2) | 829 (57.6) | 1431 (58.7) |
| | 2nd | 240 (24.0) | 396 (27.5) | 636 (26.1) |
| | 3rd | 148 (14.8) | 205 (14.2) | 353 (14.5) |
| | 4th | 10 (1.0) | 9 (0.6) | 19 (0.8) |

Abbreviations: COVID-19, coronavirus disease 2019; LC, laboratory confirmed; yrs, years. [a] Sociodemographic characteristic at the time of enrolment. [b] Participants enrolled into the reactogenicity subset were asked to report if they had any systemic events three days prior to enrolment. [c] Product mixing refers to participants who received more than one vaccine brand.

dose is provided in S3 Table. There were 2440 observations among the 2439 enrolled participants; 2294 (94.0%) were through Day 42 (264 person years observed) and 2183 (89.5%) through week 13, the end of follow up (S4 Table). In the reactogenicity sub-study, 956 (95.6%) out of the enrolled 1000 participants completed the 7-day daily follow-up period, with daily response rates ranging from 874 of 956 (91.4%) to 893 of 956 (93.4%; S5 Table).

### Solicited systemic reactogenicity events

Of the 956 participants in the reactogenicity subset who completed follow up, 595 (62.2%; CI = 0.59-0.65) reported one or more systemic reactogenicity events. Among the 479 participants (50.1% of 956) who reported more than one event, the

median number of events was 4 (interquartile range 2–6 events). The three most frequently reported events were fatigue (422; 44.1%), headache (370; 38.7%) and malaise (346; 36.2%). The median duration of each individual event ranged from 1 to 2 days (S6 Table). The proportion of severe systemic reactogenicity events ranged from 2.3% (22) among those reporting nausea to 5% (48) among those reporting malaise (S7 Table). The distribution of the individual events based on their severity is shown in Fig 3. Except for headache, the frequency of systemic reactogenicity events was significantly higher during the first two days of vaccination (and on the third day for fatigue) than prior to vaccination (S5 Table).

Sex and vaccine brand were significantly associated with the solicited systemic events (Table 2). The odds of experiencing ≥1 systemic reactogenicity events were higher among nonpregnant females than among males (adjusted odds ratio [aOR]=1.81; 95% CI 1.28-2.55; p=0.001) as well as among pregnant females than among males (aOR=1.69; 95% CI 1.03-2.78; p=0.039). The odds of reporting the events were approximately two-fold higher among Johnson & Johnson recipients than among Pfizer recipients (aOR 2.05; CI=1.40-3.00; p<0.001) while they were approximately four-fold higher among Moderna recipients than among Pfizer recipients (aOR 4.19; CI 2.34-7.51; p<0.001). In the event-specific analyses, Pfizer vaccine was significantly associated with lower odds of each individual systemic reactogenicity event than Johnson & Johnson and Moderna vaccines. Compared to male participants, female participants had higher odds of nausea and fatigue, while non-pregnant females had higher odds of malaise, chills, and headache (S8–S15 Tables).

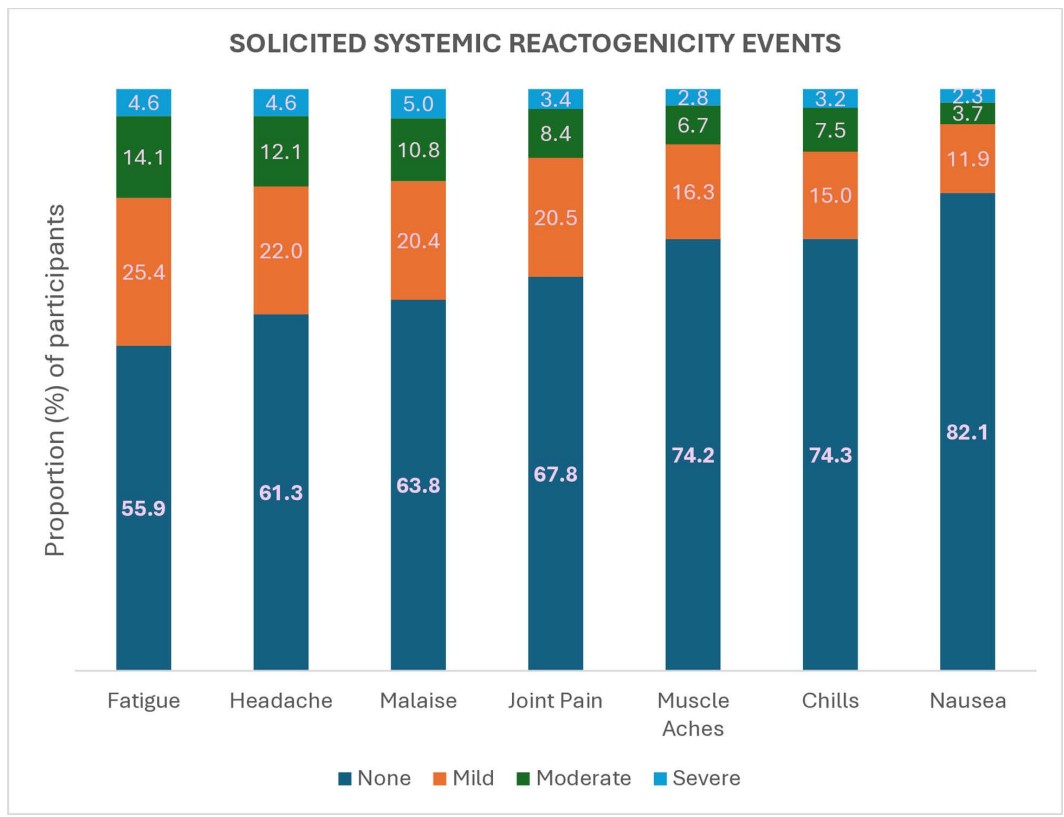

**Fig 3. Distribution of solicited systemic reactogenicity events by severity.** None denotes participants who did not report any systemic reactogenicity event. Fever was not ranked by severity since its severity was not solicited during the follow up period. The severity of the systemic reactogenicity events was determined based on their ability to interfere with the normal daily activities of the participants. Mild events did not interfere with the normal daily activities of the participants. Moderate events somewhat interfered with the normal daily activities of the participants. Severe events were considerable and prevented the normal daily activities of the participants.

**Table 2. Analysis of factors associated with solicited systemic reactogenicity events.**

| Baseline sociodemographic characteristic | | Any systemic events [c] | | Univariate analysis | | | Multivariate analysis [a] | | |
|---|---|---|---|---|---|---|---|---|---|
| | | n [e] | % (95% CI) | Odds ratio | 95% CI | p-value [b] | Odds ratio | 95% CI | p-value [b] |
| Age | 17-39yrs | 421/672 | 62.7 (58.9-66.3) | Ref | .. | .. | Ref | .. | .. |
| | 40-59yrs | 137/216 | 63.4 (56.6-69.8) | 1.03 | (0.75-1.42) | 0.837 | 0.92 | (0.63-1.33) | 0.652 |
| | 60＋yrs | 37/68 | 54.4 (41.9-66.5) | 0.71 | (0.43-1.18) | 0.184 | 0.61 | (0.34-1.07) | 0.086 |
| Sex | Male | 124/223 | 55.6 (48.8-62.2) | Ref | .. | .. | Ref | .. | .. |
| | Female, not pregnant | 351/523 | 67.1 (62.9-71.1) | 1.63 | (1.18-2.25) | **0.003** | 1.83 | (1.30-2.59) | **0.001** |
| | Female, pregnant | 120/210 | 57.1 (50.1-63.9) | 1.06 | (0.73-1.56) | 0.747 | 1.69 | (1.02-2.77) | **0.040** |
| Dose | 1 dose | 347/573 | 60.6 (56.4-64.6) | Ref | .. | .. | Ref | .. | .. |
| | 2 doses, no product mixing [d] | 59/101 | 58.4 (48.2-68.1) | 0.91 | (0.59-1.41) | 0.685 | 0.97 | (0.62-1.52) | 0.890 |
| | 2 doses, product mixing [d] | 81/127 | 63.8 (54.8-72.1) | 1.15 | (0.77-1.71) | 0.501 | 1.10 | (0.71-1.71) | 0.668 |
| | 3 doses, no product mixing [d] | 23/30 | 76.7 (57.7-90.1) | 2.14 | (0.90-5.07) | 0.084 | 2.68 | (1.09-6.57) | **0.032** |
| | 3 doses, product mixing [d] | 78/116 | 67.2 (57.9-75.7) | 1.34 | (0.88-2.04) | 0.178 | 1.21 | (0.77-1.92) | 0.409 |
| | 4 doses, product mixing [d] | 7/9 | 77.8 (40.0-97.2) | 2.28 | (0.47-11.07) | 0.307 | 2.23 | (0.41-12.01) | 0.352 |
| Brand | Pfizer | 197/364 | 54.1 (48.8-59.3) | Ref | .. | .. | Ref | .. | .. |
| | Johnson & Johnson | 316/492 | 64.2 (59.8-68.5) | 1.52 | (1.15-2.01) | **0.003** | 2.05 | (1.40-3.00) | **<0.001** |
| | Moderna | 82/100 | 82.0 (73.0-89.0) | 3.86 | (2.23-6.69) | **<0.001** | 4.13 | (2.30-7.42) | **<0.001** |
| Comorbidity | No | 430/691 | 62.2 (58.5-65.9) | Ref | .. | .. | Ref | .. | .. |
| | Chronic respiratory disease or asthma | 25/37 | 67.6 (50.2-82.0) | 1.26 | (0.62-2.56) | 0.514 | 1.24 | (0.60-2.60) | 0.560 |
| | Chronic heart disease | 6/10 | 60.0 (26.2-87.8) | 0.91 | (0.25-3.26) | 0.885 | 1.09 | (0.29-4.18) | 0.896 |
| | Chronic renal disease | 3/3 | 100 (29.2-100) | .. | .. | .. | .. | .. | .. |
| | Diabetes | 4/6 | 66.7 (22.3-95.7) | 1.21 | (0.22-6.67) | 0.824 | 1.11 | (0.19-6.44) | 0.908 |
| | Immunocompromised/ immunosuppressed | 106/179 | 59.2 (51.6-66.5) | 0.88 | (0.63-1.23) | 0.461 | 0.86 | (0.58-1.28) | 0.458 |
| | Allergy | 21/30 | 70.0 (50.6-85.3) | 1.42 | (0.64-3.14) | 0.391 | 1.21 | (0.53-2.76) | 0.654 |

Abbreviations: CI, confidence interval; yrs, years. Logistic regression model was used for both univariate and multivariate analysis. [a] Adjusted for all variables in the table. [b] P<0.05 was considered statistically significant. [c] Any systemic events includes fever, headache, chills, fatigue, joint pains, muscle aches, malaise, and nausea. [d] Product mixing refers to participants who received more than one vaccine brand. The total number of participants was 956. [e] n denotes the number of participants who had any systemic events.

## Non-reactogenicity events

A total of 164 post-vaccination hospitalization events were reported by the participants during the follow-up period. The distribution of these events was as follows: 133 (81.6%) were due to delivery at term, 14 (8.5% of non-reactogenicity events and 2.6% [95% CI 1.4-3.5%] of 535 pregnancies) were due to pregnancy related complications (S16 Table), three (1.8%) were due to elective surgery (S17 Table), and 14 (8.6%) were due to other medical conditions (S18 Table). Four of the 14 (28.6%) non-pregnancy, non-elective hospitalizations occurred within 42 days of vaccination (i.e., vomiting, high blood pressure, dengue fever, and generalized malaise; S18 Table) but were not consistent with any of the adverse events of special interest (AESI) prioritized for COVID-19 safety surveillance [15].

Six other non-reactogenicity events which did not result in hospitalization were reported by study participants (S19 Table). Four of these six reported AEFI occurred among participants receiving first dose Johnson & Johnson vaccine three to 52 days post-vaccination and included reduced breast milk production in one participant, breast swelling in another, and itchiness of the left lower limb in two participants. The itchiness was associated with numbness in one case and with blistering in another. Another AEFI of joint pain, dizziness and stoppage of menses occurring on the day of vaccination was reported among an individual receiving Johnson & Johnson vaccine as a first booster dose. One AEFI consisting of

headache and heart palpitations two days after receipt of Pfizer vaccine as a first booster was also reported. All of these non-hospitalization AEFI resolved without sequelae except in one individual who had ongoing numbness of the left lower limb at the end of follow up.

## SMS response rate

A total of 814 weekly follow up messages were sent via SMS to 90 participants. The overall response rates were as follows: invalid responses were submitted for 35.0% (285 of 814) of the messages, complete appropriate responses were submitted for 34.4% (280 of 814), there were no responses to 29.0% (236 of 814), and partial responses were submitted for 1.6% (13 of 814). The distribution of the SMS response rates as per the week of follow up is outlined in S20 Table.

## Discussion

We conducted a post-authorization active VSS study to enhance the understanding of the nature of adverse events following COVID-19 vaccination in Kenya. Few post-authorization COVID-19 active VSS activities were conducted in sub-Saharan Africa. In a landscape review, the WHO Africa region conducted only 23 of 543 active safety surveillance activities for COVID-19 globally, the lowest among WHO regions [22]. We found that solicited systemic reactogenicity events were common; most study participants (62.2%) reported one or more events, though most of these were non-severe and lasted for one to two days. Studies done in other settings have also demonstrated that vaccine-related adverse events were frequent, mainly mild to moderate and well tolerated by the participants [23]. A cross-sectional study carried out among Ghanaian healthcare workers in both public and private health settings between 16th March and 5th May 2021 showed that about 80% of COVID-19 vaccine recipients experienced at least one systemic reactogenicity event, a slightly higher proportion than in our study, with the events lasting between zero and two days, a similar duration to that observed in our study [24]. Fatigue and headache were the most commonly reported systemic reactogenicity events by our study participants. This was also observed in COVID-19 vaccine safety evaluation studies conducted in Ethiopia and Ghana [24,25]. Moreover, a study conducted in the US to evaluate the safety profile of COVID-19 vaccines also observed that headache was the most reported systemic reactogenicity event though fatigue was less frequently reported in that study [26]. However, we found that the frequency of headache in the post-vaccine period was not significantly higher than in the three days prior to vaccination. No serious solicited systemic reactogenicity event was reported during our study.

Sex was significantly associated with experiencing the solicited systemic reactogenicity events. Female participants had higher odds of experiencing the solicited systemic reactogenicity events than male participants. This observation mirrored the findings of cross-sectional studies conducted in Uganda and Saudi Arabia [27,28]. Pregnant women comprised approximately 25% of the study sample, providing an opportunity to generate evidence in this population which was excluded from COVID-19 pre-authorization trials. We observed that there was no differential risk for systemic reactogenicity events among pregnant women. A study conducted in the US also reported that both pregnant and non-pregnant women had similar adverse events profile [29].

Vaccine brand was also strongly linked to the likelihood of solicited systemic reactogenicity events. The multivariate analysis we performed demonstrated that Pfizer vaccine was associated with the lowest odds of reactogenicity while Moderna vaccine had the highest odds among the products evaluated. These differences in the reactogenicity among vaccine brands have been reported in multiple studies, [30,31] including an active safety surveillance study conducted in the US which found that that Pfizer vaccine was the least reactogenic while Moderna was the most reactogenic [32]. This heterogeneity in reactogenicity possibly arises from the technology employed during vaccine manufacture which includes inherent biological differences among these vaccines (mRNA versus vector vaccines), inclusion of different adjuvants and varied dosage [33].

The prevalence of pregnancy complications within our cohort (3%) was comparable to that in the general population within our and other similar settings. The prevalence of spontaneous abortion among women of reproductive age in Kenya

was 10% in 2022, [34] and the prevalence of pre-term birth in sub-Saharan Africa has been estimated at 3–49% [35]. The background rates of preterm delivery and spontaneous abortion in other settings are similar, ranging from 3% to 22% [36]. No other adverse events of special interest (AESI) for COVID-19 vaccines [36] were identified during the study. However, the final sample size was much smaller than the target sample size. It was sufficient to detect events occurring at a frequency of ≥1,136 per 100,000 person years within a 42-day risk window, yet most AESIs are rare, occurring at a much lower frequency [36]. Nevertheless, the sample size was sufficient to rule out a ≥ 2-fold increase in the risk of acute aseptic arthritis following COVID-19 vaccination assuming background rates at the upper range of 1000–1600 per 100,000 person years or a ≥ 3-fold increase assuming a background rate of 500 per 100,000 person years [20,21].

The SMS response rate in our study was very low (34.4%) in comparison to a study carried out in Australia to assess the performance of automated text messages to monitor adverse events following immunization in general practice which had a response rate of 72.6% [37]. The low acceptability of the SMS-based active VSS was also observed in a study done in Zimbabwe (31% of the enrolled participants responded), pointing to the challenges experienced in low-and middle-income countries which hinder compliance such as low education level, unemployment and low access to phone and phone credit. However, this Zimbabwean study also demonstrated that detection of AEFIs using SMS-based surveillance exceeded the reporting by passive surveillance despite the low compliance rates to SMS responses [38]. Therefore, there is need to adopt cost-effective digital approaches to enhance active community-based VSS in LMICs.

However, our study had some limitations. We were unable to attain the target sample size due to several constraints. Multiple and lengthy ethical, regulatory, and other approval processes which have been described previously [39] delayed study start-up yet the study had a fixed timeframe. By the time the approvals were in place, vaccination uptake had slowed down and vaccination rates remained low thereafter. At the same time, there were vaccine stocks outs during the enrollment phase of the study. In addition, there was likely selection bias as access to a mobile phone for study follow-up was an eligibility criterion; mobile phone ownership, which was recently estimated at 56% among men and 46% among women in Kilifi County, [40] is likely to be associated with socioeconomic status as affordability is a key determinant of mobile phone ownership [41]. Another limitation was that 20% of potential participants did not enroll into the study for various reasons; while this enrollment rate is comparable to those of other research studies in this setting, it may nevertheless have biased our findings if the risk, profile, or reporting likelihood for AEFI may have been different in those who did not enroll. The small sample impacted our power to detect rare events as described earlier. The post vaccination events were self-reported, possibly introducing bias if participants tended to mis-, under- or over-report certain types of events. Only three of the 164 hospitalized participants provided patient-held records, therefore we largely did not have access to ICD diagnosis codes, laboratory results, or imaging results to improve the diagnostic certainty of the reported events. Children were left out of the study and relatively few individuals above 60 years of age were recruited, limiting the generalizability of the study findings to these sub-groups of the population. About 10% of the cohort was lost to follow-up, which may have affected our conclusions if the losses were associated with a different adverse events profile to those who completed follow-up. Despite these limitations, our findings remained consistent with other COVID-19 vaccine safety studies as described earlier.

Our study had the following strengths. We were able to demonstrate the feasibility of active VSS in a resource limited setting, following a standard protocol, [15] with high retention through 42 days and 13 weeks of follow up. We predominantly enrolled females of childbearing age who were left out of the initial vaccination phase due to safety concerns hence providing valuable safety data in this special sub-group of the population. Our reporting of COVID-19 vaccine related adverse events through PvERS contributed significantly to the national passive safety surveillance system. Lastly, through logistic regression analysis, we were able to adjust for important possible confounders of the association between COVID-19 vaccination and adverse events.

In conclusion, though COVID-19 vaccines had inherent adverse reactions, most of these events were non-severe and transient, lasting few days. Further, there was no increase in the risk of adverse pregnancy outcomes following COVID-19

vaccination. In the future, a similar model – whereby a discrete active VSS activity is implemented alongside the existing passive VSS – can be employed to generate evidence following new vaccine introductions or to address vaccine safety concerns. However, any future active VSS studies will need to employ measures to mitigate implementation challenges that could threaten the timeliness, completeness and representativeness of the surveillance.

## Supporting information

**S1 Table. Sample sizes needed to rule out various levels of an increase in the risk of adverse events of special interest (AESI) if no event is observed within a 42-day risk window.** Cells with a sample size of ≤10,000 are shaded. The calculation of the 95% CI if no event is observed is based on the methods proposed by Eypasch et al., 1995.
(DOCX)

**S2 Table. Margins of error for different levels of reactogenicity prevalence for a sample size of 1,000.** Margins of error are based on the width of the exact binomial 95% confidence intervals for a proportion.
(DOCX)

**S3 Table. Distribution of vaccines received by brand and dose among enrolled participants at baseline (this excludes one re-enrolled participant with follow up over two doses).**
(DOCX)

**S4 Table. Summary of follow up periods within the cohort.**
(DOCX)

**S5 Table. Frequency of systemic reactogenicity at baseline (within three days prior to vaccination) and daily during the first week post-vaccination.** P-values compare the frequency of the event post-vaccination to baseline. Day 0 is the day of vaccination.
(DOCX)

**S6 Table. Summary of systemic reactogenicity events reported within the first week of COVID-19 vaccination.**
(DOCX)

**S7 Table. Stratification of systemic reactogenicity events by severity.**
(DOCX)

**S8 Table. Analysis of factors associated with fever.**
(DOCX)

**S9 Table. Analysis of factors associated with nausea.**
(DOCX)

**S10 Table. Analysis of factors associated with malaise.**
(DOCX)

**S11 Table. Analysis of factors associated with chills.**
(DOCX)

**S12 Table. Analysis of factors associated with headache.**
(DOCX)

**S13 Table. Analysis of factors associated with joint pain.**
(DOCX)

**S14 Table. Analysis of factors associated with muscle aches.**
(DOCX)

**S15 Table. Analysis of factors associated with fatigue.**
(DOCX)

**S16 Table. Summary of pregnancy related post-vaccination hospitalization events within the cohort.**
(DOCX)

**S17 Table. Summary of elective surgeries reported as post-vaccination hospitalization events within the cohort.**
(DOCX)

**S18 Table. Summary of other medical conditions reported as post-vaccination hospitalization events within the cohort.**
(DOCX)

**S19 Table. Non-hospitalization events reported by study participants.**
(DOCX)

**S20 Table. Analysis of SMS response rate among a subset of non-reactogenicity participants.**
(DOCX)

**S1 Text. Study questionnaire.**
(PDF)

## Acknowledgments

We thank the study participants for volunteering to take part in this study. We thank Dr. David Mang'ong'o, for training our study staff and Kilifi County health facility staff on pharmacovigilance and COVID-19 vaccine safety reporting via PvERS. We thank the Kilifi County Health Management Team (CHMT), Kilifi North Subcounty Health Management Team, Kilifi South Subcounty Health Management Team, Ganze Subcounty Health Management Team, and the participating health facility in-charges for supporting the study. We thank the following KEMRI-Wellcome Trust Research Programme teams: the fieldwork team, information technology team, and the community engagement team. This manuscript has been published with the permission of the Director, Kenya Medical Research Institute.

## Author contributions

**Conceptualization:** Shirine Voller, Anthony O. Etyang, John Anthony Gerard Scott, Ambrose Agweyu, Eunice Wangeci Kagucia.

**Data curation:** Don B. Odhiambo, Boniface Karia, Eunice Wangeci Kagucia.

**Formal analysis:** Don B. Odhiambo, Boniface Karia, Makobu Kimani, Samuel Sang, Antipa Sigilai, Anthony O. Etyang, Eunice Wangeci Kagucia.

**Funding acquisition:** Anthony O. Etyang, John Anthony Gerard Scott, Ambrose Agweyu, Eunice Wangeci Kagucia.

**Investigation:** Donald Akech, Eunice Wangeci Kagucia.

**Methodology:** Anthony O. Etyang, Eunice Wangeci Kagucia.

**Project administration:** Don B. Odhiambo, Donald Akech, Ambrose Agweyu, Eunice Wangeci Kagucia.

**Software:** Boniface Karia.

**Supervision:** Don B. Odhiambo, Makobu Kimani, Ambrose Agweyu, Eunice Wangeci Kagucia.

**Writing – original draft:** Don B. Odhiambo, Eunice Wangeci Kagucia.

**Writing – review & editing:** Don B. Odhiambo, Donald Akech, Boniface Karia, Makobu Kimani, Samuel Sang, Antipa Sigilai, Shirine Voller, Christine Mataza, David Mang'ong'o, Rose Jalang'o, Martha Mandale, Anthony O. Etyang, John Anthony Gerard Scott, Ambrose Agweyu, Eunice Wangeci Kagucia.

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
