## [Decision Letter · Decision Letter 0]

21 May 2025

PGPH-D-25-00625

Active vaccine safety surveillance: Experience from a prospective cohort event monitoring study of COVID-19 vaccines in Kenya

Dear Don,

Thank you for submitting your manuscript to PLOS Global Public Health. After careful consideration, we feel that it has merit but does not fully meet PLOS Global Public Health’s publication criteria as it currently stands. Therefore, we invite you to submit a revised version of the manuscript that addresses the points raised during the review process.

We look forward to receiving your revised manuscript.

Kind regards,

Collins Otieno Asweto, PhD

Academic Editor

Journal Requirements:

1. Figure 1: please (a) provide a direct link to the base layer of the map (i.e., the country or region border shape) and ensure this is also included in the figure legend; and (b) provide a link to the terms of use / license information for the base layer image or shapefile. We cannot publish proprietary or copyrighted maps (e.g. Google Maps, Mapquest) and the terms of use for your map base layer must be compatible with our CC-BY 4.0 license.

Reviewers' comments:

Reviewer's Responses to Questions

**Comments to the Author**

1. Does this manuscript meet PLOS Global Public Health’s publication criteria?

Reviewer #1: Partly

Reviewer #2: Yes

2. Has the statistical analysis been performed appropriately and rigorously?

Reviewer #1: No

Reviewer #2: Yes

3. Have the authors made all data underlying the findings in their manuscript fully available (please refer to the Data Availability Statement at the start of the manuscript PDF file)?

Reviewer #1: Yes

Reviewer #2: Yes

4. Is the manuscript presented in an intelligible fashion and written in standard English?

Reviewer #1: Yes

Reviewer #2: No

Reviewer #1: The types of vaccines for COVID-19 were not separately studied.

The exclusion criteria were not clearly outlined, as the symptoms that were supposed to be due to side effects of the vaccines were also common symptoms of many diseases. the time of onset for the symptoms was not consistent for all participants.

The authors' methods of analysing the data were not justifiable. For example, multivariate logistic regression analysis might not be a good statistical test for variables with two outcomes.

Reviewer #2: The paper is well thought out and the manuscript is well written. However, there are a few minor issues that need to be corrected and fixed.

1. Firstly is it possible for the authors to outline the various levels of severity of the systemic events for the primary outcome as was used to determine their ability to interfere with the normal daily routines. I presume these must have had levels. If so, it is essential to outline the various levels of severity for the primary outcome as was determined by the authors.

2. There is no clear reason or justification provided for the re-enrollment of the participant who was then followed up over two COVID-19 vaccine doses. Since they were not in the reactogenicity subset for either follow-up this makes their inclusion significantly different from the other participant. Even if their inclusion had negligible effects on overall findings, why was this done and what was the potential consequences on the overall results having in mind that one participant even thought treated independently is not sufficient for a conclusion and their inclusion raises issues of bias as they were characteristically different from the rest of the sample.

3. Is it possible to add the implications of the limitations of the study to the overall findings. Is it possible to indicate to what extent the limitations affected the findings?

**Do you want your identity to be public for this peer review?** For information about this choice, including consent withdrawal, please see our Privacy Policy

Reviewer #1: **Yes: ** Kumlachew Mergiaw Abtew

Reviewer #2: No

---

## [Decision Letter · Decision Letter 1]

1 Aug 2025

Active vaccine safety surveillance: Experience from a prospective cohort event monitoring study of COVID-19 vaccines in Kenya

PGPH-D-25-00625R1

Dear Odhiambo,

We are pleased to inform you that your manuscript 'Active vaccine safety surveillance: Experience from a prospective cohort event monitoring study of COVID-19 vaccines in Kenya' has been provisionally accepted for publication in PLOS Global Public Health.

Best regards,

Julia Robinson

Executive Editor

Reviewer Comments (if any, and for reference):

Reviewer's Responses to Questions

**Comments to the Author**

Reviewer #1: All comments have been addressed

Reviewer #2: All comments have been addressed

publication criteria?

Reviewer #1: Partly

Reviewer #2: Yes

3. Has the statistical analysis been performed appropriately and rigorously?

Reviewer #1: N/A

Reviewer #2: Yes

4. Have the authors made all data underlying the findings in their manuscript fully available (please refer to the Data Availability Statement at the start of the manuscript PDF file)?

Reviewer #1: Yes

Reviewer #2: Yes

5. Is the manuscript presented in an intelligible fashion and written in standard English?

Reviewer #1: Yes

Reviewer #2: Yes

Reviewer #1: The authors have made some revisions to the manuscript.

Reviewer #2: All issues have been addressed and i have no more issues with the research article.

**Do you want your identity to be public for this peer review?** For information about this choice, including consent withdrawal, please see our Privacy Policy

Reviewer #1: No

Reviewer #2: No
